

# Regional-scale groundwater analysis with dimensionality reduction

Márk Somogyvári[1,2], Fabio Brill[2,1], Mikhail Tsypin[3,4], Lisa Rihm[1,2] and Tobias Krueger[1,2]

[1]Integrative Research Institute on Transformations of Human-Environment Systems (IRI THESys), Humboldt-Universität zu Berlin, Berlin, Germany

[2]Geography Department, Humboldt-Universität zu Berlin, Berlin, Germany

[3]GFZ German Research Centre for Geosciences, Geosystems Department, Section 4.5 Subsurface Process Modeling, Potsdam, Germany

[4]Institute of Applied Geosciences, Technische Universität Berlin, Berlin, Germany

*Correspondence to*: Márk Somogyvári (mark.somogyvari@hu-berlin.de)

**Abstract.** Given the importance of groundwater for freshwater provision and groundwater-dependent ecosystems, understanding climate effects on groundwater changes at a regional scale is essential. In this paper, we propose a new way of applying dimensionality reduction for such purpose, not over the collected data, nor over any calibrated models, but over the misfits between the modeled and observed groundwater levels. This methodology highlights local differences in climate-groundwater relations and can be used to identify regions with different vulnerabilities in a data-driven way.

The approach takes gridded groundwater level data and gridded precipitation and evapotranspiration data as input. Linear water balance models are set up for each grid cell in an independent way. The misfits between the water balance model simulations and groundwater levels are used for the dimensionality reduction-based regionalization, with which areas of different groundwater behavior are identified.

We demonstrate the potential of our methodology in the Berlin-Brandenburg region, Germany, where groundwater is a major
freshwater source at risk. We show that groundwater level changes are linearly related to climatic variations at a monthly scale, even in areas with strong anthropogenic influences. The dimensionality reduction further reveals an approximate regionalization of groundwater behavior, which can be used as a basis for more detailed investigations.

**Key findings**

- The climate-groundwater relations in the lowland area of Berlin-Brandenburg can be estimated with linear models
- Dimensionality reduction methods can identify regions with temporal discrepancies in the expected groundwater dynamics

## 1 Introduction

Groundwater levels are increasingly an issue of climate change impacts and sustainable resource management (Ashraf et al.,
2017; Chávez García Silva et al., 2024; Taylor et al., 2013; Wu et al., 2020). Groundwater resources are essential for freshwater supply to human settlements, food production, and many industrial activities. Shallow aquifers can further be related to lakes, rivers, and wetlands, thus supporting critical ecological functions (Kløve et al., 2011). Acute or imminent shortage of



groundwater (real or perceived), e.g. during a phase of drought, can therefore trigger a wide range of cascading impacts, and has already been a source of conflicts, even in countries like Germany (Sodoge et al., 2024).

As groundwater systems are changing, different areas experience different transitions: decreasing and increasing groundwater levels could happen in different parts of the same region, driven by changes in natural or anthropogenic forcings, or a combination thereof, at different spatiotemporal scales. Therefore, groundwater needs to be managed differently in different parts of the same groundwater system, which leaves us with the scientific task of disentangling the complexity of overlapping processes in a region.

We can address this issue via regionalization, i.e. the identification of areas where the groundwater dynamics act in a similar way. In hydrology, regionalization methods are traditionally used to approach ungauged or scarcely-gauged catchments, by transferring knowledge from similarly behaving catchments (Guo et al., 2021). The accurate modeling of surface-runoff requires a lot of data from hydrology and weather stations. As most catchments in the world are without such infrastructure, regionalization methods make it possible to apply hydrological models in such catchments that were calibrated in similar

catchments with stations available. Our study takes inspiration from this approach but with some major modifications.

In the spatial context of Central Europe, and the German state of Brandenburg in particular, data scarcity is less of a problem. We would rather call our context knowledge-scarce, where the parameters behind the investigated hydrological processes are not well known (Somogyvári et al., 2024). The goal of our regionalization is not to transfer knowledge from one investigated area to another, but rather to conduct an intercomparison between multiple areas to identify the ones with anomalous

hydrological behavior. We apply this approach to a regional-scale groundwater context, focusing mainly on the climate-groundwater dynamics, instead of the typical application to surface-runoff modeling.

Traditional groundwater modeling approaches either focus on global/continental scale water systems (Condon et al., 2021), or on specific sites or aquifers with practical research questions (Galsa et al., 2022; Jasechko et al., 2024; Tóth et al., 2023). Assessing the groundwater behavior at a regional/mesoscale level is still relatively understudied (c.f. (Chávez García Silva et

al., 2024)), and mainly done by water authorities, while for surface water hydrology, catchment scale or higher-level analysis is the standard methodology (Flügel, 1997).

We refer to 'regional scale' as an analysis covering several hundred square kilometers, which implies that multiple groundwater catchments, multiple levels of administration, and spatial planning are included. At this spatial scale, data collection and processing are typically divided among different agencies and derived information is not yet disseminated in a consistent

manner (Matherne and Megdal, 2023). Country- or nationwide activities that affect the water cycle – such as inter-basin water transfers, new industrial sites, discontinuation of mining activities, or rewetting of drained peatlands – require a consistent picture at this scale, however. For the case at hand, local experts stated that potential solutions to water-related problems, amplified by regional climate change, are often designed too small in scale, and suffer from a lack of cooperation between political planning units (Arndt and Heiland, 2024; Uhlmann et al., 2023).





From the perspective of data availability, recent progress is promising: weather data products are becoming more and more accessible at a few kilometer grid scales, while a sufficient number of monitoring wells allows the interpolation of aquifer water tables over the same grid with minimal uncertainties (Arkoc, 2022). Investigating groundwater changes first at a regional scale is also in line with the downward model development approach (Sivapalan et al., 2003), i.e. starting with coarse simple models and further refining them and increasing their complexity along the identified issues.

One of the early solutions to approaching regional-scale groundwater systems was the use of process-based flow models. The unit basin concept of gravity-driven flow systems (Tóth, 1963) has explicitly dealt with issues of scales and showed how a nested groundwater flow system can describe groundwater behavior at a regional (basin) scale. Today, numerical flow models can operate at this basin scale to understand complex flow system dynamics (Tóth et al., 2023).

Process-based groundwater models have been widely used coupled together with catchment or regional scale hydrological
models to simulate groundwater dynamics (Dams et al., 2012; Seidenfaden et al., 2022). For example, (Jing et al., 2018) coupled the mesoscale hydrological model (mHM), with an OpenGeoSys 3-D parameterized flow model. The same approach was used to infer the effects of climate change on groundwater levels by (Jing et al., 2020). (Pujades et al., 2023) compared this approach against global models, showing how mesoscale models outperform their global counterparts. In the review of (Refsgaard et al., 2022), which focused on catchment scale models, those studies including a groundwater component all used
process-based models (Epting et al., 2018; Markstrom et al., 2008).

Regional-scale groundwater studies are also mostly based on process-based simulations (Zhou and Li, 2011). (Hellwig et al., 2020), for example, used MODFLOW to investigate drought propagation over Germany and (Tsypin et al., 2024) developed a thermal-hydraulic model in GOLEM for the region of Berlin-Brandenburg. (Amanambu et al., 2020) in a review in the context of climate change effects on groundwater called for models of higher complexity, to represent better the relevant flow
processes. (Jing et al., 2018) arrived at similar conclusions; in their opinion, simple water-balance type models cannot represent groundwater heads, especially in low-flow situations, so to properly focus on near-surface flow dynamics a 3-D process-based model is needed. They also argued that, while 3-D models work better at larger scales, data uncertainties and knowledge limitations are becoming increasingly prominent at those scales.

In contrast, data-driven and simple water balance methodologies do not aim at resolving small-scale physics and hence do not
face the issue of limited data on the relevant processes. Instead, they rely on statistical relationships between the different data at the scale of observation, though with model structures informed by theory. Water balance and data-driven models quantify the different fluxes of the flow system and use mass conservation principles (water balance) or statistical relationships (data-driven models). These models rely less on geological and environmental knowledge, but more on data quality and quantity. This is advantageous in a regional setting, where there is usually more data available than system understanding. The latter is
usually inferred via the interpretation of data – e.g. the hydrostratigraphy is characterized after interpretation of the drilling profile of a monitoring well.





Data-driven methods are very popular for surface runoff modeling where techniques such as water balance modeling (Mason et al., 1994), multilinear regression (Clarke, 1973), and machine learning methods (Nevo et al., 2022) have been widely explored. On the downside, these methods are prone to exhibiting a black-box behavior, meaning they work well for simulation

and forecasting purposes but are not necessarily designed to infer the properties of the modeled system. In a recent preprint (Ebeling et al., 2024) used the random forest technique, a machine learning approach, to predict characteristic spatial control features of groundwater drought dynamics from individual wells in Germany. Their study showcases data-driven methods as an ideal tool for analyzing large sample datasets, as they were able to analyze and cluster more than 6000 multidecade groundwater well time series systematically.

Dimensionality reduction techniques are also widely used for hydrological regionalization, for example, principal component analysis by (Gottschalk, 1985), or multidimensional scaling by (Solans and Mellado-Díaz, 2015). (Giese et al., 2020) classified groundwater level dynamics within monitoring wells similarly, applying the classification to the time series directly. (Haddad and Rahman, 2023) analyzed the relationships between the frequencies of flooding and different potential predictors by clustering multiple data-driven multilinear regression and generalized additive models.

In this paper, we are proposing a novel methodology to understand the spatial variability of groundwater dynamics using data-driven modeling. We focus on climate-groundwater relations at a regional scale, using coarse simple models as the initial step of a downward model development process. Inspired by the concept of hydrological regionalization, we show how dimensionality reduction methods can be applied in the context of climate-groundwater dynamics.

Our hypothesis is that in basins with temperate climate and periglacial geomorphology, groundwater response dynamics to

weather forcings can be modeled using linear approaches (such as water balances). Local anomalies would then be reflected in deviations from these linear relations and can be used to identify areas of strong anthropogenic influence, or special environmental/geological conditions with a different climate-groundwater relation. Hence, instead of applying dimensionality reduction to the observed data directly, or to any model parameters, we propose using such techniques over the model misfits, to infer areas that behave in an anomalous way compared to the groundwater response to weather forcing represented by the

model fit.

In the following, we will demonstrate how water balance modeling with gridded weather and groundwater data can reveal local anomalies of climate-groundwater dynamics. By delineating these anomalies, subsequent more focused studies could better define a region of interest, with similar groundwater dynamics. Due to the sheer size of the gridded water balance results, the model evaluation is aided by regionalization techniques. Hence, there is no need to look at the results cell by cell, but the

delineated areas can be treated as similar. Overall, the regional scale modeling provides insight to the general groundwater dynamics of Brandenburg in a data-driven way.



## 2 Study area and data

We have developed our methodology in the Berlin-Brandenburg region of Germany, where groundwater is a topical issue (Kuhlemann et al., 2020; Tsypin et al., 2024; Zielhofer et al., 2022).


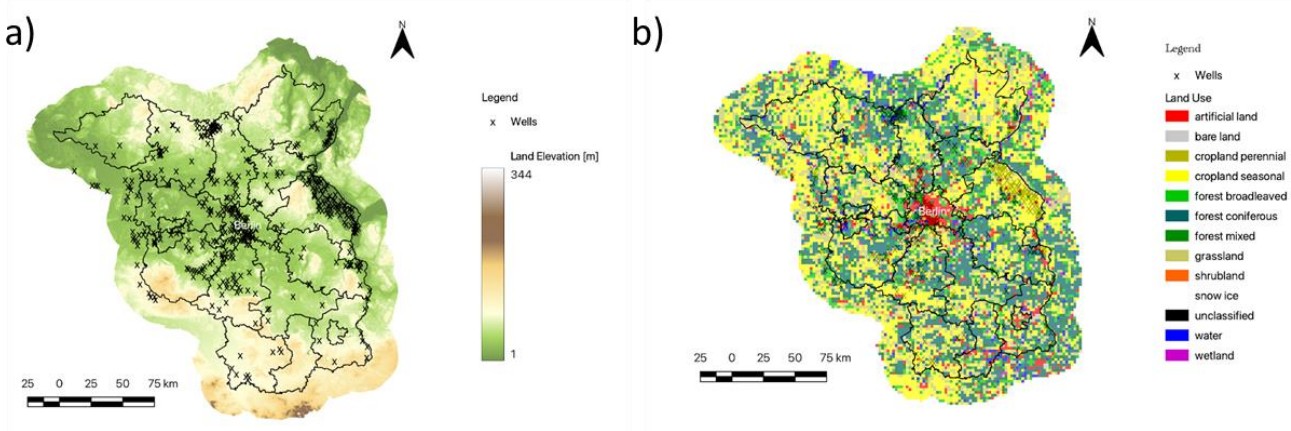

**Figure 1: Berlin-Brandenburg study area: a) Topography [source: OpenDEM] and locations of used groundwater monitoring wells [sources: Wasserportal Berlin (SenUVK, 2023), Auskunftsplattform Wasser Brandenburg (Landesamt für Umwelt Brandenburg, 2023)], b) Land cover [source: (Pflugmacher et al., 2018)]. Black lines correspond to administrative subdivisions.**

### 2.1 Brandenburg overview

Our exact study area is defined by the administrative boundary of the federal states of Brandenburg and Berlin, plus an additional 30 km wide buffer zone to include more measurement stations and avoid edge effects around the state border where possible (Fig. 1). Note that the boundary of the study area is not a natural boundary of any catchment or aquifer. The boundary is due to the data availability, as in Germany environmental data is collected at the federal state level. We will show later, that

the methodology does not rely on any natural environmental boundaries, which could be advantageous in the research context of understudied regions.

The shallow hydrogeology of the Berlin-Brandenburg region is characterized by a series of Quaternary-Tertiary aquifers of differing sand-gravel-mud proportions. According to (Limberg and Thierbach, 2002), five separate aquifers can be identified, numbered AQ1-5 (or GWL1-5 in the German literature) from top to bottom. These aquifers are not completely separated by

impermeable layers; glacial erosion has formed several highly permeable channels between them filled with sand-dominated deposits. These connections have been the focus of hydrogeological studies in the last decades, as they can lead to saltwater contamination from the Rupelian AQ5 to the main aquifer of drinking water production (AQ2).

Our study focuses on the top unconfined aquifer AQ1, which is most exposed to climatic impacts. These impacts can further propagate to the deeper aquifers due to the interlinkages. The AQ1 was formed by glacial valley deposits, larger meltwater

runoff paths, and sand end moraines. The aquifer has a variable thickness between 5 and 40 meters, and hydraulic conductivity



values between $2 - 6 \times 10^{-4} m/s$. The depth to groundwater is 1-5 meters in lowland areas and can reach up to 80 meters at higher elevations.

The regional scale groundwater flow system is controlled by the surface elevation and the topography of the underlying Rupelian aquitard (Frick et al., 2019). Recharge areas of the flow system are in the north part of Brandenburg, in the higher

elevation areas east of Berlin, and in the south part of Brandenburg. Discharge areas are located in the western part of Brandenburg, where the main Havel River leaves the state. It is important to note that in mining areas, such as the Lausitz region in the south-east of Brandenburg and neighboring states, the natural groundwater system is under extreme anthropogenic influence.

## 2.2 Weather forcing data

Climatic data for the study region is taken from the CER v2 dataset (Jänicke et al., 2017). This open dataset provides dynamically downscaled climate model data over a 2 x 2 km spatial grid at multiple timescales. The dataset was prepared specifically for the Berlin-Brandenburg region and was validated against 211 weather stations in the region. Data include monthly precipitation and monthly actual evapotranspiration, which are used here.

In a previous study (Somogyvári et al., 2024), we have successfully used this dataset to simulate surface water dynamics of a

lake near the city of Berlin. Our experience showed the advantage of a dynamically downscaled gridded dataset compared to weather data interpolated by simpler means, as the latter could not account for the high spatial variability of precipitation, which is becoming increasingly prominent during extreme events.

## 2.3 Groundwater data

Open-access groundwater data were used. In the city federal state of Berlin, groundwater data are collected by the Berliner

Senatsverwaltung für Umwelt, Mobilität, Verbraucher- und Klimaschutz (SenUMVK). In the federal state of Brandenburg, these data are collected by the Ministerium für Landwirtschaft, Umwelt und Klimaschutz (MLUK). The groundwater data were downloaded at a monthly resolution. We focus on wells that have complete/near-complete monthly time series for the period 1990-2023.

Our analysis only focuses on the actual water table, as defined by the hydraulic heads in the top groundwater body (AQ1).

Hence, the well data are further filtered, keeping only the wells that are screened at the top aquifer layer. Because this information was not available or correct everywhere, for additional control the well data are validated against the official distance-to-groundwater dataset of Brandenburg, keeping only those data where the measured groundwater levels and the distance-to-groundwater values are within 2 meters. This also filters out wells with outlier behavior, for example with strong local anthropogenic influence or wells that are part of some isolated groundwater body with an anomalous pressure regime.

With these criteria, the final number of wells used in the analysis is 504, with locations shown in Fig. 1a.



## 3 Methodology

In this section, we present our workflow for analyzing the groundwater dynamics. First, we give an overview (Fig. 2), then we show in detail the different steps and methods.

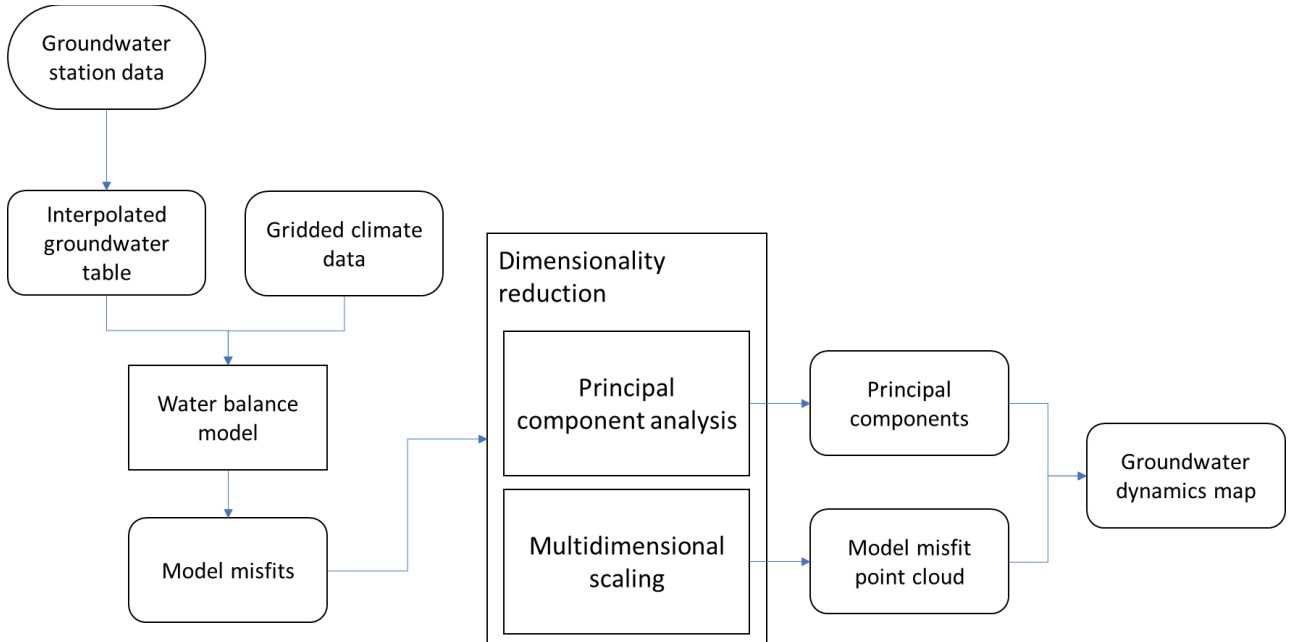

**Figure 2: Workflow for analyzing groundwater dynamics via model misfits and dimensionality reduction.**

The workflow starts with water balance modeling, where independent water balance models are set up for each grid cell in the investigation area. Subsequently, the focus is not on the water balance model itself, but on its misfits. The misfits of each grid cells are analyzed together using dimensionality reduction techniques: principal component analysis or multidimensional scaling. These methods generate a point cloud representation of the misfit relations, which can then be used to label all the
grid cells of the area to map out the different groundwater dynamics patterns. The water balance model is a simple choice of representation that will be evaluated in this paper. In principle, the workflow can involve a more complex modeling step.

### 3.1 Data interpolation

While the climate forcing data (CER v2) is in a gridded format, the water table is estimated from the well data via kriging
interpolation separately for each month within the timeframe. The kriging grid matches the underlying grid of the CER v2 dataset, with pixels of 2km x 2km. The universal kriging algorithm is used here, with a spherical variance model, implemented in the PyKrige Python package (Murphy, 2014). The kriging parametrization was selected after a grid search on possible kriging models, and we have chosen the one with the best fit with the original data. The groundwater table of each month is interpolated independently, meaning that the chosen variogram is always fitted on the data it is being used with. The



interpolation was done over a rectangular area, which was then cropped down to the study area of Brandenburg plus a 30km
buffer zone (Fig. 1).

## 3.2 Subsurface water balance

In this study, we apply the water balance modeling approach over a regional scale grid. We handle every grid cell independently

without considering any explicit cell-to-cell flows (these flows are handled implicitly). This independence will be important
in the latter parts of the analysis.

The water balance for a grid cell can be formulated as:

$$\Delta S = (WA + \Delta F) * b', \tag{1}$$

where:

$$WA = P - ET, \tag{2}$$

is water availability (sometimes also referred to as climatic water balance) (in mm/month), $P$ and $ET$ are precipitation and
actual evapotranspiration, respectively (both in mm/month). $\Delta S$ is the monthly change in storage within the pixel (in
m³/month). The conversion factor $b'$ accounts for the change in units between the terms.

The term $\Delta F$ represents all additional water flow to and out of the investigated grid cell. In most locations of the study domain,

this is the net subsurface (i.e. groundwater) inflow, but in pixels with relevant surface water in or outflows, that flow is also
considered in this term. Assuming that the changes in the ground- and surface water flow system are negligible over the
investigation timescale, the $\Delta F$ term can be assumed to be constant. The water balance can then be closed by estimating this
value.

The change in storage can be linked to the change in groundwater level by modifying the conversion factor:

$$\Delta z = (WA + \Delta F) * b, \tag{3}$$

Here, $b$ takes care of the scaling which is needed to convert the storage change in a pixel to groundwater level change. If the
unit of $\Delta z$ is taken the same as $WA$ and $\Delta F$, then $b$ is unitless. The water balance equation can be rearranged to the shape of a
linear function between the groundwater level change and the water availability:

$$\Delta z = b * WA + a, \tag{4}$$

This notation introduces parameter $a$ as the intercept of the linear model. In this paper, we use this notation exchangeability
freely, hence whenever we refer to linearity, we also mean that the water balance holds up. As mentioned above, the water
balance can be closed by calibrating the value of $\Delta F$ to make the right side of Eq. (3) fit the observed $\Delta z$ within a pixel. We
have shown that this is equivalent to the linear formula of Eq. (4). In practice, the latter was used with the Python
implementation of linear regression in the scikit-learn Python package (Pedregosa et al., 2011).





## 3.3 Time lag estimation


The groundwater does not respond to the weather forcing immediately, there is a time lag. This lag depends on multiple factors, such as the thickness of the unsaturated zone, the type of land cover and the soil moisture content. The identification of this lag is essential to set up the water balance model properly. For this reason, initially we run the water balance model with multiple time lag scenarios. In each grid cell, we chose the lag which provided the best fit with the observed data.

## 3.4 Dimensionality reduction


To further analyze the modeling results, the water balance misfits are regionalized using dimensionality reduction. A calibrated water balance model simulates the groundwater response to the dynamic climatic factors. If there are any other dynamic changes in the groundwater system not considered in the model, these would appear as anomalies in the modeled groundwater levels. A simple example of a missing term would be a change in water abstraction rates. A more complex example would be a change in land cover over time which could change the groundwater recharge. Analyzing the misfits indicates where these non-climatic factors play an important role. Because all pixels were handled independently by the water balance modeling, the correlations revealed by the dimensionality reduction will not be biased by the modeling procedure.


The misfits can be defined as:

$$misfit(x, y, t) = z_{GW}(x, y, t) - \sum_t \Delta z(x, y, t), \tag{5}$$


Note that the second term is a cumulative sum, that is used to convert the simulated water balance to actual groundwater levels. The obtained misfit is also a time series, not just a scalar value. Two examples are shown in Fig. 3.

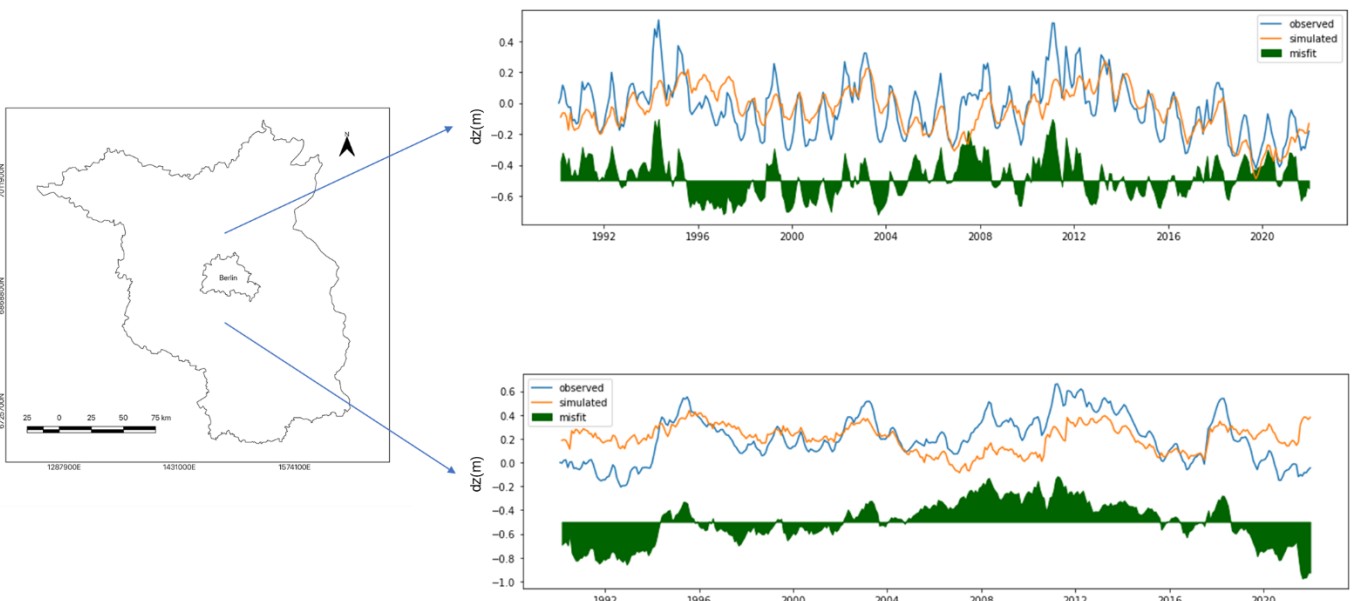

**Figure 3: Example differences between observed and modelled groundwater levels at two different locations.**



Due to the sheer number of grid cells, it is difficult to interpret or analyze all of the misfits directly. Dimensionality reduction can reduce the size of this dataset while maintaining its main features, and the patterns within. Dimensionality reduction is widely used on high-dimensional, or very large datasets to reduce the computational load of the applied analysis method, and to avoid overfitting, and it is often applied for visualization purposes (Van Der Maaten et al., 2009).

Here, the role of dimensionality reduction is to reduce the misfit array to a point cloud, where each point represents a pixel of

the investigation area. The reduction happens along the temporal axis while retaining the dominant features of the misfit time series. We show two different algorithms for dimensionality reduction: multidimensional scaling and principal component analysis. With these two examples, we would like to emphasize that dimensionality reduction can be implemented into the methodology in a flexible way, using different algorithms.

### 3.4.1 Multidimensional scaling

The first step of multidimensional scaling (MDS) is to create a so-called dissimilarity matrix (also known as distance matrix), a matrix where the value in cell i,j represents the difference between the misfit time series of the i-th and j-th grid cell on the map.

$$D[i,j] = misfit[i] - misfit[j],$$ (6)

The dissimilarity matrix is a symmetric matrix, with zero diagonal values. Multidimensional scaling projects the dissimilarity matrix into a Euclidian n-dimensional space, in the form of a point cloud, while keeping the distances between the individual points as close to the distance matrix as possible. The point cloud can then be visualized in 2-D or 3-D form.

Multiple types of MDS exist. In this study, we used the metric MDS implemented in the scikit-learn Python package (Pedregosa et al., 2011). Metric MDS uses nonlinear transformations to project the point cloud via minimizing the cost

function:

$$C = \sum_{i<j}\left(D[i,j] - r_{i,j}\right)^2,$$ (7)

where $r_{i,j}$ is the projected Euclidian distance of the i-th and j-th point within the projected point cloud.

### 3.4.2 Principal Component Analysis

Principal component analysis (PCA) is a linear dimensionality reduction method, that uses linear combinations of the original

variables, to transform the data into a new coordinate system. Principal components are constructed in a way that best represents the variance of the original data.

PCA eliminates the correlation between the variables by constructing new independent variables, the orthogonal principal components (PCs). The first PC represents the greatest portion of the variance of the original dataset, the second PC the second greatest variance while being uncorrelated with the first PC, and so on until the entire variance of the original data is explained

by new independent variables, with the bulk of the variance consolidated in the first few PCs.



We apply PCA on the flattened version of the misfit array:

$$\boldsymbol{M}[i,j] = misfit(\underline{x}_i, t_j), \tag{8}$$

where $\underline{x}_i$ is a one-dimensional running index for all the grid cells. Hence array $\boldsymbol{M}$ has rows equal to the total number of grid cells and columns equal to the length of the investigated timeframe ($t$). To apply PCA to this array, first, it is transformed into the sample covariance matrix, where each element represents the covariance between the samples:

$$\boldsymbol{S} = \frac{\boldsymbol{M}^T \boldsymbol{M}}{n-1}, \tag{9}$$

where n is the total number of grid cells. The final coefficient matrix of the PCA is then calculated via singular value decomposition of the covariance matrix ($\boldsymbol{A}$). This ensures the orthogonality among the principal components. PCA performs a linear transformation of matrix $\boldsymbol{M}$ using the coefficient matrix $\boldsymbol{A}$:

$$\boldsymbol{Z} = \boldsymbol{M}\boldsymbol{A}, \tag{10}$$

Which results in a transformed $\boldsymbol{Z}$ matrix, where each column represents an independent principal component. For dimensionality reduction, matrix $\boldsymbol{Z}$ gets truncated, in the case of our study keeping only the first three columns ($\boldsymbol{Z}_T$). The final truncated $\boldsymbol{Z}_T$ matrix can be plotted as a 3-D point cloud. In our analysis, we used the scikit-learn implementation of the PCA algorithm (Pedregosa et al., 2011).

**3.5 Coloring/labeling/clustering**

Dimensionality reduction methods create point-cloud representations of the data, in which each point represents a cell of the original spatial grid. The point cloud dimensions can be used directly to define a linear color scale to plot on the map. For PCA this could result in a separate map for each principal component. Taking three dimensions of the point clouds can also be used to define an orthogonal RGB color space. This color space can be used to map out the point cloud on a single map, each pixel having an RGB value based on the point location within the point cloud. The point cloud can also be clustered in a subsequent step, to identify similar types of areas. In this study, we use a Gaussian mixture model from the scikit-learn Python package (Pedregosa et al., 2011) to cluster the point clouds. Gaussian mixtures were preferred over the other methods such as k-means because it works better with high-density point clouds with no distinct boundaries between them.

**4 Results & Discussion**

**4.1 Model fit and performance**

The water balance model was set up using a one-month time lag, meaning the precipitation and actual evapotranspiration of one month is used to calculate the groundwater level change for the next month. This time lag was selected after testing multiple possible time-lags, choosing the one with the best overall fit (see supplementary figure S1). The estimated net water outflow values ($-\Delta F$) are mapped out for the research area as shown in Fig. 4a.


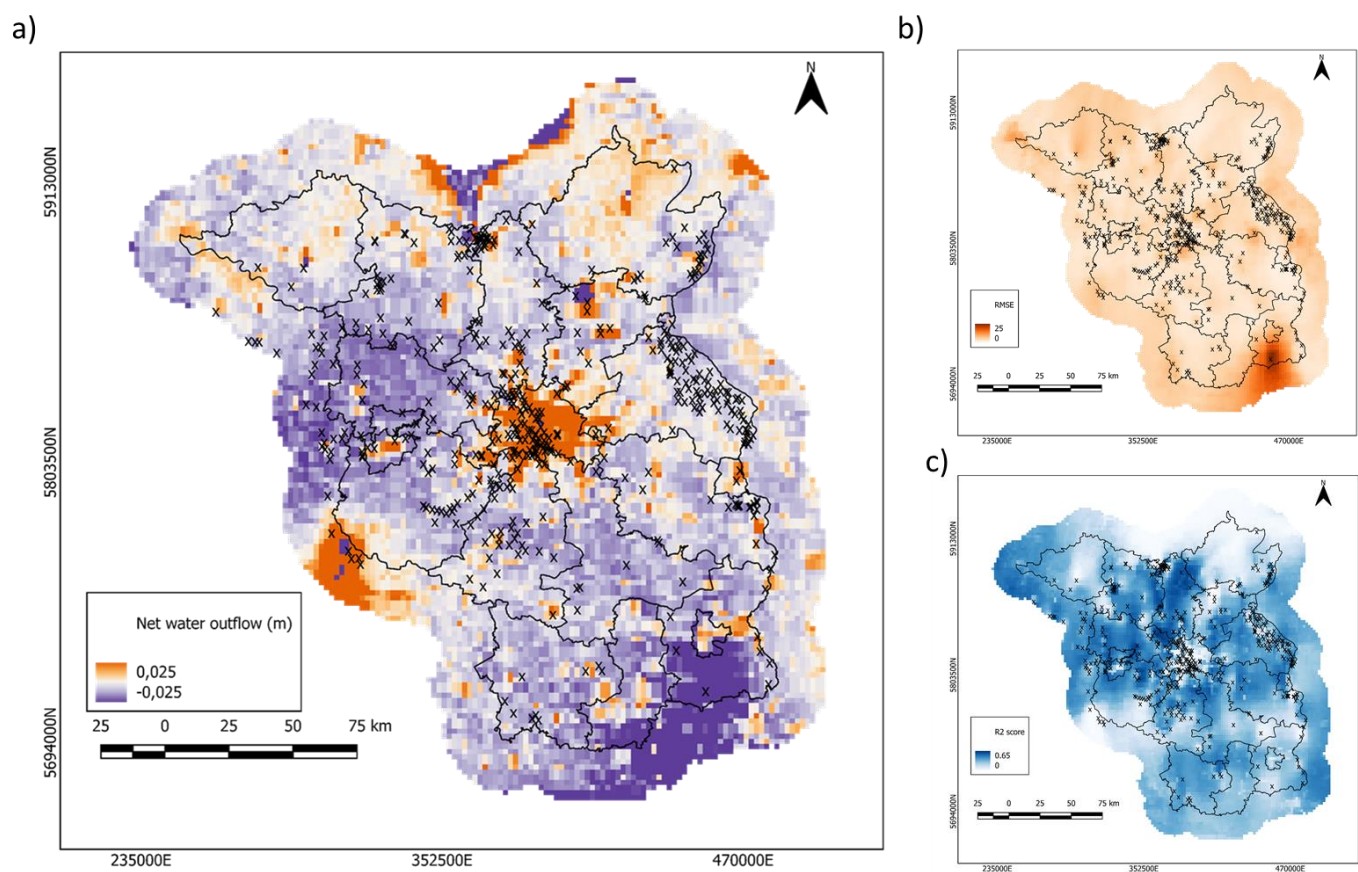

**Figure 4. a) Estimated net water outflow from the water balance model. Blue color marks areas with water flowing in (water deficit, discharge areas), orange colors mark areas with water flowing out (water surplus, recharge areas), b) RMSE misfits of the calibrated water balance model, c) r2score of the calibrated water balance model.**


Although an independent water balance model was set up for each grid cell, the resultant map shows a spatially coherent result. Large areas with similar net water outflow values dominate the region, only with some occasional single-pixel outliers.



## 4.2 Spatiotemporal patterns of modeled groundwater dynamics

At a regional scale, some correlation of net water outflow can be seen with topography (compare Fig. 1a). This is in line with what we expect from a gravity-driven groundwater system. For example, the area west of Berlin shows high water inflows. This is a lowland discharge area of the groundwater system. In contrast, the region east of Berlin is an area with hills and higher elevations, which acts as a recharge zone. A similar recharge area can be seen in the north.

   This coherence does not hold in the southeast part of Brandenburg. The topography would suggest a recharge zone here, but

the mapped values show a strong water deficit. This is a former open pit mining region, where the groundwater levels are rebounding after more than a hundred years of artificial pumping. Groundwater levels here are steadily increasing, many meters over some decades. This rebound process requires a significant amount of extra water inflow which is shown in the model.

   The coherence does not hold up either in urban areas, where we see very high outflow values (compare Fig. 1b). This is due to actual evapotranspiration values being higher than the estimates in the input dataset. The city of Berlin is the most prominent

example of this behavior, but smaller cities like Brandenburg an der Havel and Cottbus are also recognizable on the map. Cities have a lot of sealed surfaces (Fig. 1b), hence most of the water surplus in these areas leaves in the form of surface flow, and only a small portion can recharge groundwater. Hence, we cannot interpret the net water outflow term here as recharge or subsurface flow. Still, the result is relevant for the general picture of the water cycle in the region, and very well shows how cities were traditionally built in a way that channels the rainwater out as quickly as possible (for a critique, see (Jia et al.,

340    2017)).

   Another very high-outflow area is the southwest edge of the map. This area is part of the Flaeming Heath, a topographically elevated area acting as a the watershed divide between the Elbe and the Havel. The area is poor in flowing waters, with many dry valleys called locally "Rummel" (Pieper et al., 2017).

   Other extreme values beyond the boundaries of Brandenburg can be attributed to the lack of groundwater data, for example in

the northernmost parts. Note that these are not boundary effects of the model, as the method treats each pixel independently and no interactions between them are simulated, but an effect of extrapolation of the groundwater data. The model uncertainty in the production of the gridded input datasets obviously poses a limitation of this study, and caution is thus advised when interpreting spatial patterns far off the measurement locations.

   The spatial map of the RMSE (Fig. 4b) confirms the plausibility of the presented simple linear water balance model over the

whole timeframe. In most of Brandenburg, the RMSE values are low: the water balance approach holds up as the groundwater levels are linearly related to the climate forcing.

   The results, again, are very coherent in space, with similar RMSE values in nearby pixels. This is no surprise, as both input data types were coherent themselves: the CER data as a result of a spatially distributed numerical model, and the groundwater data as a result of a spatial interpolation.

The coefficient of determination ($r^2$score) between the modeled and observed groundwater dynamics shows a little bit different picture (Fig. 4c). While the RMSE has a more averaging behavior over time, the $r^2$score is more sensitive to the fit of the short-



term dynamics. Here, urban areas show a very low score indicating that the dynamic groundwater behavior is not captured well, and the groundwater response to climate is less linear.

Most high-RMSE and low-$r^2$score areas can be explained by the lack of groundwater observations (Fig 1a). Some correlation is also visible with topographic elevation, especially in the north of the domain. Increasing elevation is normally accompanied with thickening of the vadose zone. Thus the observed correlation is in line with (Lischeid et al., 2021) who showed the importance of vadose zone thickness for the climate-groundwater dynamics using data from multiple sites overlapping with this region. (Ebeling et al., 2024) also indicated distance to groundwater as the most important factor when classifying groundwater dynamics in Germany.

**4.3 Dimensionality reduction**

Standard model metrics, such as RMSE and r² scores, offer a good overall assessment of the fit of the groundwater model. To gain a more detailed understanding of the model's behavior, we applied dimensionality reduction methods to analyze the misfit time series.

To demonstrate this methodology, two specific methods are presented here: multidimensional scaling (MDS) and principal component analysis (PCA). In Figure 5, plots a and c present the point clouds created by these methods respectively. In these 3-D visualizations, each point represents a pixel from the map, and the distance between the points is representative of the differences between these two points' misfit behaviors. The colors of the points are created by assigning RGB color models to their x,y,z coordinates. The colored data points are presented as maps in Fig. 5b) and d).




a)

c)

b)

d)

**Figure 5. Dimensionality reduction: a) multidimensional scaling point cloud, b) multidimensional scaling map, c) principal component analysis point cloud, d) principal component analysis map. Rivers are shown in light-blue on the maps.**

The point clouds can be plotted over different dimensions, here for the sake of simplicity, we have chosen 3-D projections. For the MDS point cloud in Fig. 5a, the axes of the plots are virtual dimensions (sometimes called feature dimensions). These dimensions cannot be tied to anything physically (or mathematically) meaningful, their sole purpose is to define a Euclidean space where the data points can be projected – maintaining the pre-defined distances between them.

In contrast, the axes of the PCA plot in Fig. 5c, are the principle components, axes along the data where it varies the most. The two-point clouds show a lot of similarities. This is expected, due to the similar nature of the two techniques: PCA is a specific case of MDS where a linear projection is used to create the point cloud.

The point clouds are elongated along the first dimension, with multiple elongated umbilical arms, spreading away from the dense parts of the cloud. This suggests a strongly different model behavior at these points from the rest. The most significant





of such features is the pixels of the Lausitz region, which are the points located in the right arms of both point clouds. Smaller internal structures are visible as well, representing strong correlations within the data.

By transferring the RGB coloring of the point clouds to the map we see spatially coherent areas as patches of the same color. This is expected, due to the interpolated groundwater table, and the small variability of input weather data. The shape of some
of the patches resembles the river network of the region, for example, an elongated pale green shape southeast of Berlin follows the river Spree, and other light green areas are shown by sections of the Oder and Elbe.

The outer areas of the point cloud can be linked with other environmental factors. Dark blue and dark green pixels outline the Lausitz area, with strong anthropogenic impacts. Individual wells with differences in groundwater dynamics stand out on these maps very will, e.g. the red spot near the east side of the map, within the city of Frankfurt an der Oder, or the light blue spot
northwest from it, which is a well in the city of Munchenberg. The wells exhibit different behaviors due to anthropogenic impacts at the first site and the higher elevation and larger distance-to-groundwater at the second site.

The Flaeming Heath stands out on this map again with a red color, and the same can be said about green areas in the north. The results are coherent locally, but not throughout the whole region. We cannot say that the dark blue color on this map means just open pit mining, this is only true in the context of the southeast area. Instead, we can say that the response style relative to
the forcing is non-unique. This is an important limitation of this methodology, which requires further analysis before making definite conclusions about the whole water system.

## 4.4 Principal Components

In this section we take a closer look to the three PCA components, which are shown in Fig. 6. In PCA, the point cloud
dimensions have a specific meaning in relation to the variance of the data. For this dataset, the first three principal components explain 85% of the variance (42%, 25%, and 8% respectively).




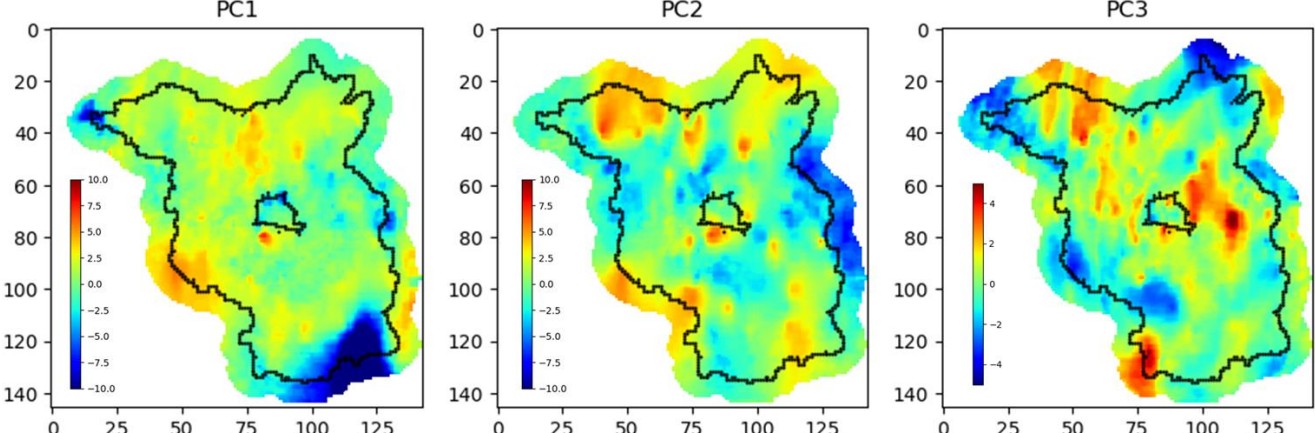

**Figure 6: Results of the principal component analysis a) 1ˢᵗ PC, b) 2ⁿᵈ PC, c) 3ʳᵈ PC.**

The first principal component highlights the main anthropogenic influences on the groundwater table. The Lausitz region in the south is very visible on this map as a negative anomaly and the city of Berlin is also in contrast with its surroundings. Another strong anomaly is visible on the northwest, this however cannot be linked to the same cause – demonstrating that non-anthropogenic effects could create similar patterns in the data. It is important to reiterate, that this is a data-driven approach, and the inferred principal components are related to data patterns, while linking these to physical processes requires interpretation.

At some locations, very small local anomalies are visible. These are caused by differences in single-well behavior. As the groundwater tables were calculated via interpolation, anomalous areas that are not sampled by enough wells are shown as such local features. A good example of this is the area east-southeast from Berlin, where only a handful of wells were used. This type of anomaly, is also visible across all principal components.

The map of the second principal component shows a lot of similarities with the topography of the region (see Fig. 1). Smaller values are visible in lowlands, like the alluvial plain of the river Oder in the east, or the valley of the Havel in the west while large values are at highlands line in the north or in the southwest.

The map of the third principal component cannot be correlated directly with environmental factors. It rather shows some similarities with the used error metric maps of RMSE and r2score. Some of the anomalies visible here highlight areas of smaller well density, for example along the region borders. The explained variance of this component is significantly less than even the second component; hence we would interpret it as the residual information remaining within the misfit dataset.

Looking at the three principal components together, we can see that in the majority of Brandenburg, the water balance misfit behavior is similar. What does this mean exactly? Together with the model error metrics, we can say that the climate-groundwater relation is mostly linear in the region and can be sufficiently modeled via water balance models. The study of (Ebeling et al., 2024) reached similar conclusions when identifying the factors underlying the groundwater dynamics in Germany. They identified meteorological forcings as the main cause, a factor that we implicitly consider in our modeling





approach. Anthropological impacts and the distance-to-groundwater were the other two major factors, but the authors also noted the importance of other local factors, such as surface waters and land use types. Anthropological impacts can be seen
here as well, as the anomalous urban and mining areas, and the distance-to-groundwater as the anomalies related to topography.

**4.5 Implications for the Berlin-Brandenburg region**

For further investigation, we look into some selected subregions in more detail (Fig. 7). First, to better delineate areas with different misfit behavior, we clustered the PCA point cloud (see supplementary figure S2), and mapped out the obtained class labels (Fig. 7a). On this map, patches of the same color represent areas with similar model misfits.

By plotting some exemplary local water balance model misfit curves, we can look into the temporality behind the groundwater regionalization obtained by the dimensionality reduction. We chose the example regions from the anomalies identified previously, with the additional criteria of having proper coverage of groundwater monitoring wells.

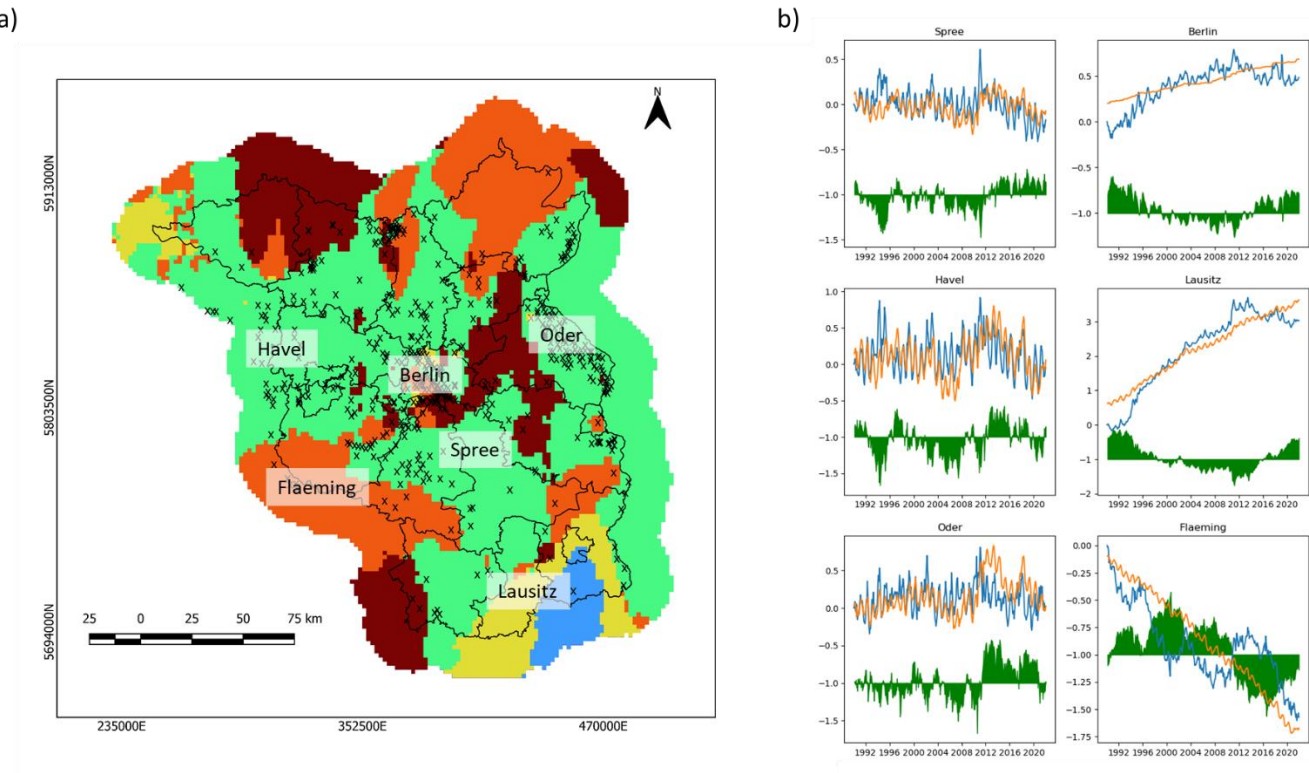


**Figure 7: Different groundwater dynamics within the Berlin-Brandenburg region: a) map projection of the clustered PCA point cloud, labeled with example locations, b) observed groundwater levels (blue), modeled groundwater levels (orange) and misfit time series (green) from the example locations.**



Three plots are selected from the catchments of the three main rivers, the Spree, the Havel, and the Oder. These alluvial plains compose the majority of the lowland areas in Brandenburg, and they show similar model fit behavior.

The observed and modeled groundwater levels show a good fit, with some very short outlier periods. These outliers are relatively minor when compared to the overall magnitude of the groundwater levels, while the increasing and decreasing trends are accurately modeled. This result again confirms that for most of Brandenburg, a linear model can capture the groundwater

dynamics well and shows that the main driver behind the groundwater dynamics is the climate.

One interesting behavior across all plots is the systematic overestimation of groundwater levels during the last 10 years of the timeframe. This is the period where all locations show a decreasing trend. The systematic model overestimation suggests a change in the groundwater response to the climatic drivers. A similar behavior was seen by (Somogyvári et al., 2024) for the decreasing levels of one groundwater-fed lake near Berlin, where the overestimation of a water balance model was explained

by changing hydrological processes.

In Berlin, the quality of the fit is worse – which is expected as the recharge processes are very different in an urban area in a way not represented by the model and input data – but the general trends and dynamics are still captured by the model. Water use in the city is also dynamically changing, with water use generally declining in the last decades, but larger building and infrastructure constructions causing temporary increases in water abstraction.

The plot from the post-mining Lausitz area shows a dramatically different picture from any other area. Here, a huge impact of the former open pit mining is reflected in a groundwater level increase of 4 meters. Interestingly, this anthropogenic impact still can be modeled by the water balance approach, and the resulting model fit is not significantly worse. This shows that the high-frequency dynamics of the groundwater are still controlled by the weather patterns and that the discontinuation of the open pit mining and, therefore, pumping can be approximated reasonably well as a constant flux within our timeframe of

investigation (1990-2022), even if the actual dynamics follow a concave trajectory.

One of the most interesting areas, that stood out at many points of our analysis was the region of the Flaeming Heath. The plot for Flaeming provides an example of a topographically elevated area, with a much thicker unsaturated zone (20m<) than other parts of Brandenburg. The dynamics here are not captured very well, the model only reconstructs the decreasing trends and the yearly dynamics, but none of the dynamics with multiyear periodicity. (Lischeid et al., 2021) suggested that areas with

thick vadose zones are more prone to climate change induced groundwater level drop and that this phenomenon may be explained by altered recharge processes. The lack of any surface water could be another explanation, the presence of the region-specific dry valleys ("Rummeln") is an indication of different hydrological processes than other areas with similar topography. However, to give an exact explanation for this anomalous behavior a more focused study for this subregion would be needed.



**5 Conclusions**

In this study, we have presented a new groundwater assessment framework, that uses data-driven models in combination with dimensionality reduction-based regionalization methods to identify areas or groundwater catchments with different characteristic behavior.

The methodology uses gridded weather and groundwater data as input, to set up independent water balance models for all grid cells of the investigation area. Dimensionality-reduction is applied over the model misfits at the pixel level to regionalize the

model fit behavior.

We have shown that the presented methodology is well applicable to regions with peri-glacial morphology under temperate (or humid continental) climate, and that it performs the best in cases of shallow-lying water table in lowlands and river valleys where groundwater is in good communication with the atmosphere. The approach does not require any detailed geological/hydrological knowledge, but rather depends on data coverage instead. This is advantageous for initial

investigations, especially as data are becoming more and more accessible. Due to the non-unique responses, however, the method poses some limitations for interpretation and instead motivates questions for further process-based investigation. Therefore, this methodology fits well into the downward model development concept of (Sivapalan et al., 2003), as an initial step before increasing model complexity.

For the Berlin-Brandenburg region, we have found that the groundwater dynamics are linearly related to the driving climate

factors. Dimensionality reduction over the model misfits helped us to regionalize areas with different characteristic behaviors. Principal component analysis showed that anthropogenic influences had the strongest impact on the non-climate-driven behavior, showing anomalies in urban and industrialized areas. The thickness of the unsaturated zone had the second strongest impact on the misfits where soil moisture effects and non-linear water transport would require more sophisticated modeling approaches. These findings show that linear models are valuable in the region for evaluating the groundwater impacts of

different climate scenarios and could even be used within forecast applications. The detailed misfit analysis with dimensionality reduction gives an approximate regionalization of groundwater behavior for the region that can be used as a basis of more detailed investigations.

The presented methodology is further extendable using non-linear modeling methods, such as machine learning techniques. (Beven, 2024) recommended to focus on model misfits of such models for knowledge production, as such models are difficult

to interpret on their own. Dimensionality reduction provides a promising addition to such an analysis in a regional setting.

**Data availability**

The CER v2 dataset is available from the website of the Chair of Climatology, TU Berlin, under the following link: https://www.tu.berlin/en/klima/research/regional-climatology/central-europe/cer (Bart et al., 2023).



The used groundwater data from Berlin, can be accessed at the water portal of the Berlin Senate: https://wasserportal.berlin.de/stationen_start.php (SenUVK, 2023). Groundwater data from Brandenburg is accessible at the water portal of Brandenburg: https://apw.brandenburg.de/ (Landesamt für Umwelt Brandenburg, 2023).

**Competing interests**

At least one of the authors is guest editor of the special issue "Current and future water-related risks in the Berlin–Brandenburg region".

**Acknowledgements**

This study was funded through the Einstein Research Unit "Climate and Water under Change" from the Einstein Foundation Berlin and Berlin University Alliance under grant no. ERU-2020-609.

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
