# Peer review of "Regional-scale groundwater analysis with dimensionality reduction"

_EGUsphere, 2024_

## Author Response (AR1)

**Response to Reviewer #1**

The manuscript by Somogyvári et al. presents an interesting approach to analyzing regional groundwater dynamics using dimensionality reduction on discrepancies between simple water balance models and observed groundwater levels. Applied to Berlin-Brandenburg, the method identifies spatial patterns in model misfits, potentially highlighting anthropogenic influences or unique hydrogeological conditions. By leveraging readily available climate and groundwater data, techniques like principal component analysis and multidimensional scaling offer insights into groundwater trends without requiring extensive prior knowledge. This data-driven approach could be valuable for assessing groundwater systems and their climate responses, helping to guide future studies.

Overall, the manuscript covers a compelling topic and presents a solid methodological framework that could appeal to a wide audience. It is well-structured and written, but I suggest the following improvements to enhance clarity and completeness:

We would like to thank the constructive criticism of the reviewer. Please find our answers and corrections below the specific comments.

Please explicitly and precisely state the clear objectives of the study in one or two sentences in the Introduction.

We added the following explanation to the Introduction:

L113 "Inspired by the concept of hydrological regionalization, we show how dimensionality reduction methods can be applied in the context of climate-groundwater dynamics. Our main objective is to delineate regions with different climate-groundwater interactions, providing a further basis for more focused modeling studies."

A clear map showing both Berlin and Brandenburg at the beginning would be helpful. In Figure 1 (and Figure 3), Berlin is labeled, but Brandenburg is not. Additionally, the white font color in Figure 1 makes it difficult to read the city name. It would also be good to ensure that figure labels are properly placed and annotations are aligned.

We have modified the maps in Fig. 1 to make them more accessible.

Minor formatting issues, such as in lines 188, 348, and 374, should also be addressed.

Will be corrected.

The study region is defined by administrative borders rather than hydrogeological ones. While practical for data availability, this could introduce biases. Could you discuss the implications of this choice?

The study region definition was chosen after data availability. As different European countries and even different German federal states are using different groundwater data platforms and standards, collection and unification of such data sets is a big

challenge. We tried to mitigate any limitation due to administrative borders by using an extra buffer zone around Brandenburg to involve more data, but indeed this could be seen as a limitation.

We would like to argue however that our proposed approach is especially designed for these situations: approaching an area with limited amount of knowledge, relying on only the data. The proposed data analysis should be able to reveal the different hydrogeological units, which is shown in the result section.

**Please see:**

L139 "We will show later, that the methodology does not rely on any natural environmental boundaries, which could be advantageous in the research context of understudied regions."

**and we modified:**

L493 "The approach does not require any detailed geological/hydrological knowledge, but rather depends on data coverage instead. The methodology can safely use data defined by administrative boundaries, without the need to exactly know any environmental borders or boundary conditions."

The descriptive discussion on aquifers AQ-1 to AQ-5 is useful, but a visual and precise representation on a map would improve clarity.

Thank you for this comment. In a classic groundwater modeling study we would agree that it is necessary to include such figure. However, because our methodology is data-driven, we don't see any major role of including such visualization beyond giving a context for the setting. We would rather just refer to other papers here, where such visualization is present.

Based on the recommendations however, we have expanded Fig. 1 with a third panel, that shows the general hydrogeological setting of the region.

Could you provide references for the hydrogeological details of AQ-1 mentioned in lines 150-152?

We included the following references into the revised manuscript:

(Limberg and Thierbach, 2002) and (Landesamt für Umwelt Brandenburg, 2023).

The study focuses on 504 filtered wells, removing the outliers and anthropogenically influenced wells, but there is no sensitivity analysis of how this affects the results. Would fewer or more wells affect the regional patterns identified? Please also provide the initial number of wells before filtering.

More than 2000 wells were considered in the beginning of the analyses from the region. The main criteria for filtering was to make sure that all used wells are screening AQ-1, so the water table for this water body can be reliably interpolated. The majority of the filtered out wells are screened at other depths.

This filtering resulted in 700 wells total. However in many cases, information about the wells were often limited to technical details, it was difficult to assess the filtered aquifer without inspecting each well individually. Therefore, we used a strict criteria of similarity between the mapped groundwater distance and the well levels. This extra filtering step resulted in the final 504 wells.

Please note that this approach does not remove anthropogenically influenced wells by design, but indeed it is prone to remove wells with irregular groundwater behavior.

**We have revised the text as:**

L178 "This method could also filter out wells with outlier behavior, for example with strong local anthropogenic influence or wells that are part of some isolated groundwater body with an anomalous pressure regime. With these criteria, the final number of wells used in the analysis got reduced to 504 (from more than 2000 originally), with locations shown in Fig. 1a."

The unit for water outflow in Figure 4 is given in (m)—is this correct?

Yes, it is a flux m3/m2 defined similarly to precipitation units (see eq. 3).

The study assumes a constant flux approximation for the groundwater recovery process in Lausitz (1990–2022), despite its non-linear nature. Could you provide further justification or discuss whether a more detailed approach would improve accuracy?

This was the intended way of using our approach.

Because the groundwater recovery in Lausitz is non-linear, it shows up on the final map results as an anomalous area. With our approach, this anomalous area can be delineated, and later closer inspected.

Technically it would be possible to replace the used linear model with more complex setups (polynomials, non-linear machine learning models), but that would be against our main intentions, as the interpretation of the misfits in this case would be more difficult (not simply the deviation from linearity). We would only recommend this as a subsequent step - following a downward development approach, with gradually increasing the complexity.

**We revised the text as:**

L508: "Keeping the underlying models linear, the observed anomalies and misfits can be linked to non-linearities in the recharge processes. The presented methodology is further extendable using non-linear modeling methods, such as machine learning

techniques. (Beven, 2024) recommended to focus on model misfits of such models for knowledge production, as such models are difficult to interpret on their own."

The manuscript refers to supplementary figure S1, but unfortunately I could not find it in the preprint.

The figure was included into the supplementary materials (separately from the manuscript file).

The manuscript states that a one-month time lag was chosen based on the "best overall fit" but does not define the criteria used for this selection. Could you please elaborate on "best overall fit"?

Best overall fit was identified using the r2score value between the two curves.

We have added this information to the revised manuscript:

L307 "This time lag was selected after testing multiple possible time-lags, choosing the one with the best overall fit using the  $r^2$  score metric (see supplementary figure S1)."

Given the variability in aquifer types (confined/unconfined) and geological settings, a constant one-month lag may not be the best choice across the entire study area. Could you discuss this from a process-based perspective rather than relying solely on statistical analysis? Since AQ-1 is the primary focus, do you think this assumption holds for other aquifers as well, or just the best results are shown here?

In the supplementary material we are showing that this assumption is valid for most of the region. It is possible to make the timelag adaptive to data, but we wanted to keep our approach simple and consistent. It would be more necessary to consider when using higher temporal resolutions (daily) or when investigating regions with larger hydrogeological variability.

In other aquifers this assumption most likely would not hold, there a different timelag (or a variable one) should be used.

The study applies universal kriging with a spherical variogram model, but details on kriging parameters and model selection are missing. Since this interpolation is critical for the entire analysis, could you elaborate on how the Kriging method and best model were chosen and discuss any uncertainties related to this approach? Did you consider the directionality in variography?

The kriging parameters were chosen using a grid search. We did not consider anisotropy in the variogram (which would technically subdivide the data used in the estimation). We have added the chosen parameters to the revised manuscript:

L207: "The universal kriging algorithm is used here, with a spherical variance model (parameters: Sill = 686.97 m, Range = 1.97 km), implemented in the PyKrige Python package (Murphy, 2014)."

We have included further discussion regarding the uncertainties of the gridded datasets into the revised manuscript:

L346 "The model uncertainty in the production of the gridded input datasets obviously poses a limitation of this study. We acknowledge that the parametrization of the kriging interpolation is a critical step in this respect. While we followed a clearly defined procedure - optimizing parameters over a grid search with respect to a performance metric in cross-validation - there is still necessarily an element of subjectivity in such approaches, e.g. in the choice of performance metric, the density of the parameter search grid, or the ratio of training-test-split. We did not account for anisotropy, which might potentially improve the results in some subregions. This being said, we believe that the uncertainty from the varying density of wells is more relevant than the uncertainty from the parametrization, and caution is thus advised when interpreting spatial patterns far off the measurement locations."

A simple linear model for the water balance has been used throughout the paper. Do you think such an oversimplification can capture the dynamic behavior of groundwater systems, especially in cases with complex hydrogeological processes, in urban areas with anthropogenic influences and in the era of climate change?

Our concept is based on using simple models, to identify areas where non-linearities play an important role. The goal was not to create an accurate data-driven simulation model, but to evaluate how the observed groundwater dynamics behave compared to a simple linear climate-groundwater situation.

The advantage of this framing is that the anomalies can be explained by non-linearities, which then can be further interpreted by different processes.

The proposed framework however is flexible, and allows the inclusion of more complex models (such as polynomial regression or machine learning models). The interpretation of anomalies in such a setup however would be more difficult.

We have added the following to the revised manuscript:

L505 "The detailed misfit analysis with dimensionality reduction gives an approximate regionalization of groundwater behavior for the region that can be used as a basis of more detailed investigations. Keeping the underlying models linear, the observed anomalies and misfits can be linked to non-linearities in the recharge processes. The presented methodology is further extendable using non-linear modeling methods, such as machine learning techniques."

**References:**

Beven, K.: A brief history of information and disinformation in hydrological data and the impact on the evaluation of hydrological models, Hydrol. Sci. J., 69, 519–527, https://doi.org/10.1080/02626667.2024.2332616, 2024.

Landesamt für Umwelt Brandenburg: Auskunftsplattform Wasser (APW), 2023.

Limberg, A. and Thierbach, J.: Hydrostratigrafie von Berlin-Korrelation mit dem Norddeutschen Gliederungsschema, Brand. Geowiss Beitr, 9, 2, 2002.

**Response to Reviewer #2**

The manuscript of Somogyvári et al. presents a methodology for investigating groundwater dynamics at the regional level, applying dimensionality reduction on the differences between a set of groundwater level observations in the Berlin-Brandenburg region (Germany) and the respective levels simulated by a simple water balance model. Though using a subsurface water model based on the physically-based concept of mass balance, the approach is classified as data-driven since it needs several observations, which makes the method hardly applicable in data-scarce areas.

The methodology presented is sound overall, and the results are clear enough. They highlight that linear models can be applied to describe the response of the groundwater dynamics to the driving climate input for most of the analysed region. At the same time, the areas needing further studies are detected, even though their dynamics are not disentangled.

Overall, the manuscript represents an interesting methodological contribution to regionalscale groundwater analysis in regions with good data availability. It can be published after the following minor comments are addressed.

We would like to thank the reviewer for the constructive comments. We agree that good data availability is essential for the presented methodology. Please find our answers to the specific comments below.

L54: "Assessing the groundwater behavior at a regional/mesoscale level is still relatively understudied". Even though the authors already added a reference, this statement could be more strongly justified.

We have modified this section as:

L54 "The groundwater dynamics at a regional/mesoscale level is still relatively understudied (c.f. (Chávez García Silva et al., 2024)). This usually requires good understanding of interconnections between deep and shallow aquifers, while water authorities, who are responsible for characterising and monitoring catchment hydrology, only focus on surface waters and "productive" part of the hydrogeological system, i.e. freshwater aquifers, in isolation.(Flügel, 1997)"

Figure 1: I believe that this figure needs to be completely revisited. In the text, especially in the Results and Discussion section, several toponyms are used, which make it very difficult for the reader to follow the thread of reasoning. All these toponyms must be shown clearly in a much more prominent and clearer Figure 1a. In addition, a third panel should be added (let's say, figure 1c), with appropriate geolithological information related to the hydrogeology of the region analysed.

Thank you for this comment. We revised this figure following the recommendations of both reviewers.

Section 2.1: It is explained that five separate aquifers at different depths can be identified in the analysed region. The study addresses only the top unconfined one. It is unclear if/how the methodology adopted could also be used for the lower aquifers, presumably less affected by climate drivers. Please explain and discuss.

The proposed methodology is flexible and easily adaptable to different groundwater bodies. For this, only two main modifications would be needed:

- 1. Finding all the wells that screen e.g. the second aquifer, and then interpolate a water table for that aquifer.
- 2. recalculation of the climate-groundwater lag times, as it would take longer for the climate to influence the deeper aquifer layers.

The challenge of this is to identify wells from a specific aquifer body, and to obtain a proper coverage. Choosing the top unconfined aquifer was an obvious choice by us as (i) it has the most wells and (ii) we can use the groundwater distance as a criteria to identify the correct wells better.

**We have added to the text:**

L174 "Our analysis only focuses on the actual water table, as defined by the hydraulic heads in the top groundwater body (AQ1), but it would be applicable in other selected aquifers as well given enough data is available. To remove timeseries from other groundwater bodies, the well data are further filtered, keeping only the wells that are screened at the top aquifer layer."

Section 3.1 (and, consequently, Section 4): It is unclear how much the kriging algorithm (i.e., universal kriging with spherical semivariogram) influences the results achieved, especially for the zones with fewer observations. Please explain and discuss.

The chosen kriging method and parameters were selected using a grid search, to create the best possible interpolation. Still, the heterogeneous data distribution in the region have a strong impact on the uncertainty of the results. We have added further discussion in this matter on the revised manuscript at:

L346 "The model uncertainty in the production of the gridded input datasets obviously poses a limitation of this study. We acknowledge that the parametrization of the kriging interpolation is a critical step in this respect. While we followed a clearly defined procedure - optimizing parameters over a grid search with respect to a performance metric in cross-validation - there is still necessarily an element of subjectivity in such approaches, e.g. in the choice of performance metric, the density of the parameter search grid, or the ratio of training-test-split. We did not account for anisotropy, which might potentially improve the results in some subregions. This being said, we believe that the uncertainty from the varying density of wells is more relevant than the uncertainty from the parametrization, and caution is thus advised when interpreting spatial patterns far off the measurement locations"

Section 4.1: Looking at Figure S1, one can quickly agree that a one-month time lag is the most suitable in general (even though I would have used a discrete color bar rather than continuous since only integer values are in the map). However, some zones diverge from this general rule. It is unclear if the method could consider different time lags for different zones in the same study area.

The method is adaptable to different timelags. The map from S1 can be used to inform the data-driven model on which timelag to use. In case of big variability, a location specific timelag can also be assigned, but in this paper we aimed for consistency using a single timelag. Using an adaptable timelag would require additional discussions in the misfit analyses, as it could introduce artefacts in the result maps (especially when working with monthly resolution data). (please see also our reply to Reviewer #1 on using the approach in different aquifers)

Fig. 5: The maps are difficult to interpret, and the colors are quite confusing, I think, particularly for color-blind people.

We agree with the reviewer that these maps are relatively complicated to interpret. However their goal is mainly just method visualization, not necessarily analysis - for that we would like to refer to the later figures.

The RGB colorspace was chosen here because it could provide colorcoding well for the 3-dimensional point clouds, that could be mapped over to the map visualization. As there are limited number of available 3-D colorspaces, our only real alternative would have been the visualization along the individual axes. This is shown for the PCA in Fig. 6.

Clustering was also chosen as another solution for visualization and result analysis (see Fig. 7).

We noted the complexity of this figure at the end of this section:

L401 "The visualizations in Fig. 5b and d, could be difficult to interpret, due to the specificalities of the RGB visualization. Hence in the next sections we provide further solutions to visualize and analyze the dimensionality reduction results."

L417: "...the city of Berlin is also in contrast with its surroundings". Unclear. If I understand, the city of Berlin lies within the borders of the inner polygon, and I can see that the predominant color is green, as in much of the region.

We revised the text as:

L416 "The Lausitz region in the south is very visible on this map as a negative anomaly and similar behavior (although in much smaller extent) can be seen in the city of Berlin."

**References:**

Chávez García Silva, R., Reinecke, R., Copty, N. K., Barry, D. A., Heggy, E., Labat, D., Roggero, P. P., Borchardt, D., Rode, M., Gómez-Hernández, J. J., and Jomaa, S.: Multi-decadal groundwater observations reveal surprisingly stable levels in southwestern Europe, Commun. Earth Environ., 5, 387, https://doi.org/10.1038/s43247-024-01554-w, 2024.

Flügel, W.-A.: Combining GIS with regional hydrological modelling using hydrological response units (HRUs): An application from Germany, Math. Comput. Simul., 43, 297–304, https://doi.org/10.1016/S0378-4754(97)00013-X, 1997.

---

## Editor Decision (ED1)

**Minor comments on manuscript EGUSPHERE-2024-4031**

Title: Regional-scale groundwater analysis with dimensionality reduction

I would like to thank the authors for their efforts in revising and improving the manuscript in line with the referees' requests. The manuscript has now reached a satisfactory level from a scientific standpoint. However, I kindly ask the authors to pay attention to the visual presentation of the figures before publication. Please address the following points:

- Figure 1: Improve readability; legend text is too small.
- Figure 4: Improve readability; coordinates and legends in panels b and c are not readable.
- Figure 5: Improve readability; axis labels and text are too small.
- Figure 6: Improve readability; axis and labels are too small.
- Figure 7b: Improve readability; axis and labels are too small.

---

## Author Response (AR2)

Dear Dr. Alencar,

Thank you very much for your constructive comments. We have revised our manuscript accordingly.

---

## Author Response (AR3)

Dear Dr. Alencar,

Thank you very much for your constructive comments. Please see our point-by-point corrections below:

Figure 1: Improve readability; legend text is too small.

We have increased the text in the legend:

Figure 4: Improve readability; coordinates and legends in panels b and c are not readable.

We have increased the text of the legend in b) and c), and removed the coordinates from the frame:

Figure 5: Improve readability; axis labels and text are too small.

We have increased the text size throughout the figure:

Figure 6: Improve readability; axis and labels are too small.

We have modified the colorbars for better readability.

Figure 7b: Improve readability; axis and labels are too small.

We have increased the size of the text in the figure.